# Multiple cullin-associated E3 ligases regulate cyclin D1 protein stability

**Ke Lu[1†], Ming Zhang[2†], Guizheng Wei[1], Guozhi Xiao[3], Liping Tong[1], Di Chen[1]\***

[1]Research Center for Computer-aided Drug Discovery, Chinese Academy of Sciences, Shenzhen, China; [2]Department of Oncology, Johns Hopkins University, Baltimore, United States; [3]Department of Biochemistry, Southern University of Science and Technology, Shenzhen, China

**Abstract** Cyclin D1 is a key regulator of cell cycle progression, which forms a complex with CDK4/6 to regulate G1/S transition during cell cycle progression. Cyclin D1 has been recognized as an oncogene since it was upregulated in several different types of cancers. It is known that the post-translational regulation of cyclin D1 is controlled by ubiquitination/proteasome degradation system in a phosphorylation-dependent manner. Several cullin-associated F-box E3 ligases have been shown to regulate cyclin D1 degradation; however, it is not known if additional cullin-associated E3 ligases participate in the regulation of cyclin D1 protein stability. In this study, we have screened an siRNA library containing siRNAs specific for 154 ligase subunits, including F-box, SOCS, BTB-containing proteins, and DDB proteins. We found that multiple cullin-associated E3 ligases regulate cyclin D1 activity, including Keap1, DDB2, and WSB2. We found that these E3 ligases interact with cyclin D1, regulate cyclin D1 ubiquitination and proteasome degradation in a phosphorylation-dependent manner. These E3 ligases also control cell cycle progression and cell proliferation through regulation of cyclin D1 protein stability. Our study provides novel insights into the regulatory mechanisms of cyclin D1 protein stability and function.

## Editor's evaluation

This fundamental study advances our understanding of the role of cullin-associated E3 ligases in regulating cyclin D1 protein stability. The evidence supporting the conclusion is compelling, from siRNA screens and ectopic expression approaches. This paper is of potential interest for cell biologists studying the mechanisms of protein posttranslational modification.

**\*For correspondence:**
di.chen@siat.ac.cn

[†]These authors contributed equally to this work

## Introduction

Cyclin D1 is a key factor controlling cell cycle progression. It forms a complex with CDK4/6 and functions as a regulatory subunit in G1/S phase transition during cell cycle progression. Proteasome degradation is one of the critical post-translational regulatory mechanisms modulating steady-state protein levels of cyclin D1 during normal cell cycle progression (*Qie and Diehl, 2020*). Cullin-Ring complexes comprise the largest known family of ubiquitin ligases. Human cells express seven different cullins, CUL1, -2, -3, -4A, -4B, -5, and -7, and each of them nucleates a different ubiquitin ligase (*Petroski and Deshaies, 2005*). Recent studies demonstrate that F-box proteins Fbxw8, Fbx4, Fbxo31, and AMBRA1 interact with CUL1/7 and mediate cyclin D1 degradation in a phosphorylation-dependent manner (*Chaikovsky et al., 2021*; *Kumar et al., 2005*; *Li et al., 2018*; *Maiani et al., 2021*; *Okabe et al., 2006*; *Santra et al., 2009*; *Simoneschi et al., 2021*). However, it is not known if other E3 ligase(s) is also involved in maintaining steady-state protein levels of cyclin D1. In this study, we have screened an E3 ligase siRNA library and identified three additional cullin-associated E3 ligases, which

mediate cyclin D1 ubiquitination and proteasome degradation. Our findings indicate multiple cullin-associated E3 ligases participate in the regulation of cyclin D1 stability in the cells.

## Results and discussion

To determine if cullins are required for cyclin D1 degradation, CUL1–7 (CUL1, -2, -3, -4A, -4B, -5, and -7) expression plasmids were co-transfected with cyclin D1 into HEK293 cells. Forced expression of CUL1–7 significantly suppressed cyclin D1 protein levels (*Figure 1—figure supplement 1A*). In contrast, the levels of cyclin B1 (G2/M regulator) and cyclin A (S phase regulator) were not affected (*Figure 1—figure supplement 1A*), indicating that the CUL1–7 specifically regulate cyclin D1 protein levels. Consistent with these findings, RNAi-mediated knockdown of *cullins* significantly enhanced levels of endogenous cyclin D1 (*Figure 1—figure supplement 1B*). The knockdown efficiency of these siRNAs to each *cullin* mRNA was about 60–90% (*Figure 1—figure supplement 2*). In addition, the silencing specificity of each *cullin* siRNA has been verified through both western blotting and real-time PCR assays (*Figure 1—figure supplements 3 and 4*). Results of luciferase assay also showed that cullins inhibited cyclin D1 activity (*Figure 1—figure supplement 5*). In contrast, silencing of the cullins led to increased cyclin D1 activity (*Figure 1—figure supplement 6*).

We then determined whether cullin-induced cyclin D1 degradation is ubiquitin-dependent and performed ubiquitination assay. We found that forced expression of CUL1–7 resulted in polyubiquitination of wild-type (WT) cyclin D1 (*Figure 1—figure supplement 7*). It was noted that CUL4B and CUL7 exhibited the most significant effect on cyclin D1 ubiquitination compared with other cullins. In contrast, CUL1–7 had no effects on the ubiquitination of mutant cyclin D1 with phosphorylation defect (T286A) (*Figure 1—figure supplement 7*). It has been reported that Thr286 phosphorylation is required for cyclin D1 ubiquitination (*Diehl et al., 1997*). Our results suggest that cullin-induced cyclin D1 ubiquitination is phosphorylation-dependent. A proteasome degradation inhibitor, MG132, was used to further determine if the cullins-mediated cyclin D1 degradation is through a proteasome-dependent mechanism. MG132 significantly reversed cyclin D1 degradation induced by CUL1–7 (*Figure 1—figure supplement 8*). Compared with their effects on cyclin D1 protein degradation, CUL1–7 had no significant effects on cyclin D1 mRNA expression (*Figure 1—figure supplement 9*). These results demonstrated that these seven human cullins may play critical and redundant roles in cyclin D1 degradation through a phosphorylation-dependent mechanism.

Each cullin protein interacts with different multi-subunit ubiquitin ligases (*Petroski and Deshaies, 2005*). Hundreds of substrate receptor subunits associated with different cullin-Ring ligases have been characterized in mammalian cells (*Petroski and Deshaies, 2005*). To identify additional E3 ligases which are associated with cullin proteins and are involved in cyclin D1 degradation, we generated an siRNA library targeting known cullin-associated proteins. This library contains triplicated siRNAs targeting to specific genes encoding for 154 ligase subunits, including F-box, SOCS, BTB-containing proteins, and DDB proteins. We transfected these siRNAs with 3xE2F-Luc reporter construct into NIH3T3 cells. 3xE2F-Luc reporter specifically responds to cyclin D1 stimulation (*Ohtani et al., 1995*). After screening the siRNA library, we found that 3xE2F luciferase activity was enhanced by 24 E3 ligases (>1.5-fold increase) (*Figure 1—figure supplement 10*). These results suggest that multiple E3 ligases associated with different cullin proteins are involved in cyclin D1 degradation. We selected five E3 ligases, which showed highest stimulation activities on the 3xE2F-Luc reporter and found that all of them have the ability to reduce cyclin D1 protein levels in HEK293 cells (*Figure 1—figure supplement 11*). Among these E3 ligase subunits, Fbxw8 (specific for CUL1 and -7) has been reported to induce cyclin D1 degradation (*Okabe et al., 2006*). The roles of other E3 ligase subunits, such as Keap1 (associated with CUL3), DDB2 (associated with CUL4A and -4B), and WSB2 (associated with CUL2 and -5) in cyclin D1 degradation have not been reported.

To further test the hypothesis that cyclin D1 degradation is mediated by multiple E3 ligases, we determined if different E3 ligases, Keap1, DDB2, and WSB2, are involved in cyclin D1 proteolysis. We first examined the interaction of these E3 ligases with cyclin D1 through co-immunoprecipitation (co-IP) assays. HEK293 cells were transiently transfected with HA-tagged Keap1 and treated with MG132 to prevent cyclin D1 degradation. Endogenous cyclin D1 was detected in the Keap1 immunoprecipitate, and this interaction was enhanced by the CUL3 (*Figure 1A*). Similarly, the interaction of DDB2 with endogenous cyclin D1 was enhanced by the CUL4A and -4B (*Figure 1B*). Consistent with these results, we also observed that CUL2 and -5 enhanced the interaction between WSB2 and endogenous

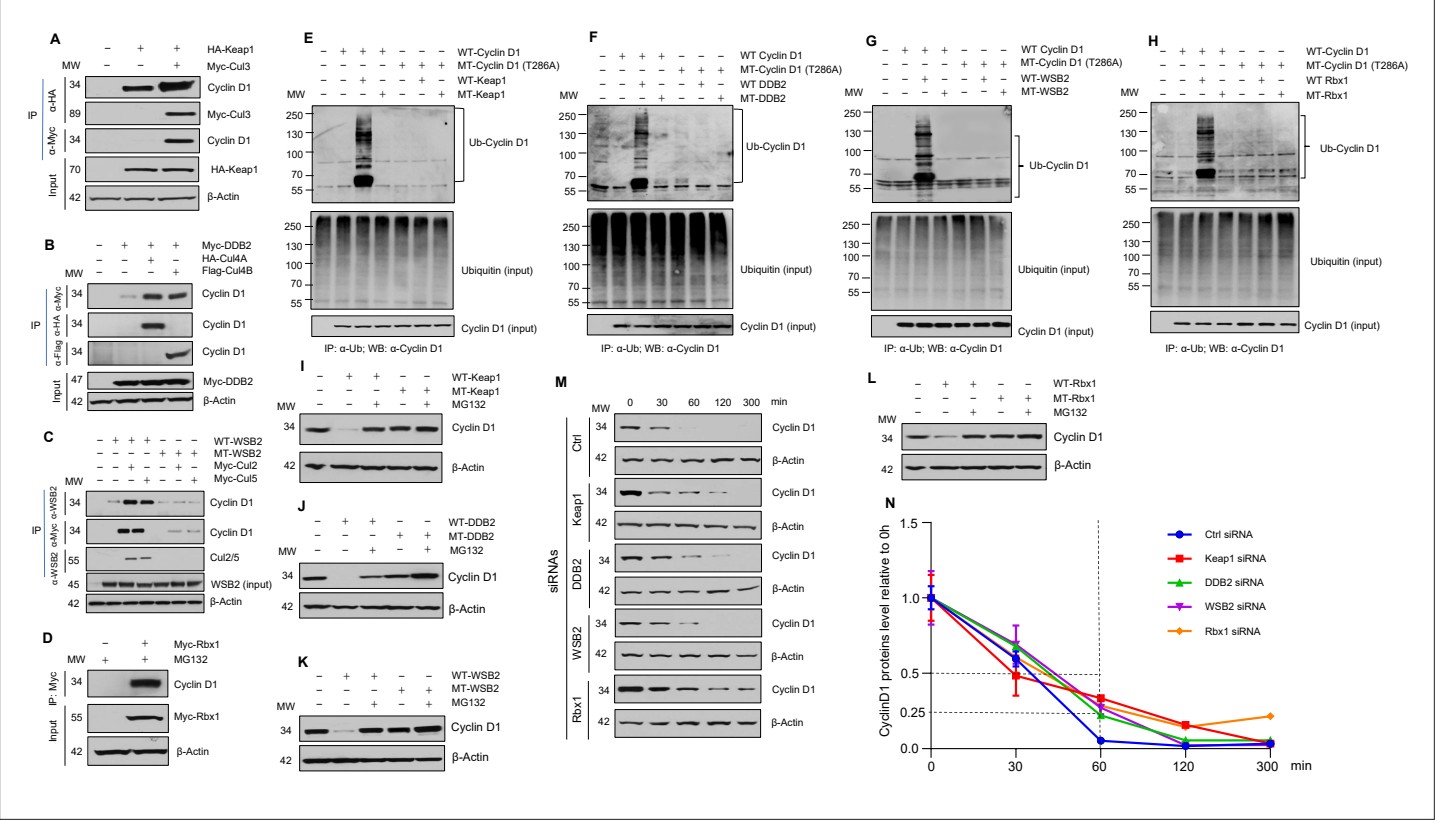

**Figure 1.** Cullin-associated E3 ligases mediate cyclin D1 ubiquitination and proteasome degradation. (**A**) Co-immunoprecipitation (co-IP) of Keap1, CUL3 with endogenous cyclin D1. HA-Keap1 and Myc-CUL3 were co-transfected into HEK293 cells with the MG132 treatment (10 µM, 4 hr incubation). 24 hr after transfection, the cell lysates were collected. To detect interaction of Keap1 with cyclin D1 or CUL3, co-IP was performed using the anti-HA (α-HA) antibody followed by the western blot using the anti-cyclin D1 or anti-Myc (α-Myc) antibody. To detect the interaction between CUL3 with cyclin D1, co-IP assay was performed using the anti-Myc antibody followed by the western blot using the anti-cyclin D1 antibody. (**B**) Co-IP of DDB2 and CUL4A/4B with endogenous cyclin D1. Myc-DDB2, HA-CUL4A, and Flag-CUL4B were transfected into HEK293 cells with the MG132 treatment (10 µM, 4 hr incubation). Co-IP was performed using the anti-Flag (α-Flag), anti-HA, or anti-Myc antibody followed by the western blot using the anti-cyclin D1 antibody. (**C**) Co-IP of WSB2 and CUL2/5 with endogenous cyclin D1. Wild-type (WT) or mutant form of WSB2 (SOCSΔ364–400) were co-transfected with Myc-CUL2/5 into HEK293 cells with MG132 treatment (10 µM, 4 hr incubation). Co-IP was performed using the anti-Myc or anti-WSB2 (α-WSB2) antibodies followed by the western blot using the anti-cyclin D1 antibody. (**D**) Co-IP of Rbx1 with endogenous cyclin D1. Myc-Rbx1 was transfected into HEK293 cells with MG132 treatment (10 µM, 4 hr incubation). Co-IP assay was performed using the anti-WSB2 or anti-Myc antibody followed by the western blot using the anti-cyclin D1 antibody. (**E–H**) Ubiquitination assay. WT or mutant cyclin D1 (T286A) were co-transfected with WT or mutant Keap1, DDB2, WSB2, or Rbx1 expression plasmids into HEK293 cells with the treatment of MG132 (10 µM, 4 hr incubation). Co-IP was performed using the anti-Ub (α-Ub) antibody followed by the western blot using the anti-cyclin D1 (α-cyclin D1) antibody. (**I–L**) WT or mutant Keap1, DDB2, WSB2, and Rbx1 expression plasmids were co-transfected with cyclin D1 expression plasmid into HEK293 cells with the MG132 treatment (10 µM, 4 hr incubation). Cyclin D1 protein levels were detected by the western blot analysis. (**M, N**) Protein decay assay. HEK293 cells were transfected with scramble siRNA (Ctrl) or Keap1, DDB2, WSB2, or Rbx1 siRNA. The cell lysates were collected 0, 30, 60, 120, or 300 min after cycloheximide treatment (80 µg/ml) and the cyclin D1 protein levels were detected by the western blot analysis and were quantified (n=3).

The online version of this article includes the following source data and figure supplement(s) for figure 1:

**Source data 1.** Numerical data obtained during experiments represented in *Figure 1*.

**Source data 2.** Original western blot files for *Figure 1*.

**Figure supplement 1.** Cyclin D1 degradation is mediated by multiple cullins.

**Figure supplement 2.** Silencing efficiency of the cullin siRNAs was determined by the luciferase assay.

**Figure supplement 3.** Specificity of cullin 4A and 4B siRNAs.

**Figure supplement 4.** Specificity of cullin siRNAs.

**Figure supplement 5.** Cullins affect cyclin D1 activity.

**Figure supplement 6.** Silencing of cullins enhances cyclin D1 activity.

**Figure supplement 7.** Cyclin D1 ubiquitination is mediated by multiple cullins.

*Figure 1 continued on next page*

*Figure 1 continued*

**Figure supplement 8.** Proteasomal inhibitor reverses cullin-mediated cyclin D1 degradation.

**Figure supplement 9.** Effects of cullins on cyclin D1 mRNA expression.

**Figure supplement 11.** Five different cullin-associated ubiquitin ligases mediate cyclin D1 degradation.

**Figure supplement 12.** Silencing effects of *Keap1, DDB2, WSB2,* and *Rbx1* siRNAs are specific to their target genes.

cyclin D1 (*Figure 1C*). Rbx1 is associated with CUL1–7 and was used as a control in this study. The results of co-IP assay showed that Rbx1 interacted with endogenous cyclin D1 (*Figure 1D*).

We then examined the effects of Keap1, DDB2, and WSB2 on cyclin D1 ubiquitination and degradation and used loss-of-function mutants of these E3 ligases to do experiments. In addition, we used Rbx1 as a control in these experiments. For Keap1, we used BTB domain deletion form which abolishes its binding with CUL3 (*Furukawa and Xiong, 2005*). For DDB2, we used the WD motif deletion form which blocks its binding to its substrates (*Nag et al., 2001*). For WSB2, we used the C-terminal deleted mutant which loses its binding to CUL2 and -5. And for Rbx1, we used the mutant form Rbx1$^{C53A/C56A}$ which dramatically reduced the ligase activity (*Ohta et al., 1999*). We found that WT but not mutant Keap1 (F-box mutation) induced cyclin D1 ubiquitination (*Figure 1E*). In contrast, Keap1 had no effect on the ubiquitination of phosphorylation mutant cyclin D1 (T286A) (*Figure 1E*). Similarly, WT DDB2, WSB2, and Rbx1 induced cyclin D1 ubiquitination and mutant DDB2, WSB2, and Rbx1 had no effects on cyclin D1 ubiquitination (*Figure 1F–H*). In addition, Keap1, DDB2, WSB2, or Rbx1 had no effects on the ubiquitination of phosphorylation mutant cyclin D1 (T286A) (*Figure 1E–H*). These results indicate that Keap1, DDB2, and WSB2 interact with cyclin D1 and mediate cyclin D1 ubiquitination in a phosphorylation-dependent manner. We then determined the effects of these E3 ligases on cyclin D1 degradation by western blot analysis. Keap1, DDB2, WSB2, and Rbx1 induced WT but not T286A mutant cyclin D1 degradation (*Figure 1I–L*). In contrast, mutant Keap1, DDB2, WSB2, or Rbx1 had no effect on cyclin D1 degradation (*Figure 1I–L*). Consistent with these findings, silencing of Keap1, DDB2, WSB2, or Rbx1 resulted in the stabilization of cyclin D1 protein and significantly prolonged the half-life of cyclin D1 (*Figure 1M and N*). The silencing specificity of siRNA for each E3 ligase subunit has been verified through real-time PCR assays (*Figure 1—figure supplement 12*). To determine the effects of these E3 ligases on protein levels of endogenous cyclin D1, we transfected Keap1, DDB2, WSB2, and Rbx1 siRNA into human colon cancer cell line HCT-116 cells and found that knocking down of *Keap1, DDB2, WSB2,* or *Rbx1* significantly enhanced endogenous cyclin D1 protein levels (*Figure 2A, D, G, and J*). Taken together, these findings indicate that cyclin D1 degradation is mediated by multiple E3 ligases which are associated with different cullin proteins.

We then determined whether these E3 ligases affect the function of cyclin D1 during normal cell cycle progression. Over-expression of Keap1, DDB2, WSB2, or Rbx1 markedly inhibited the growth of HCT-116 cells. In contrast, these E3 ligases had no effect on cell proliferation in the cells stably transfected with T286A mutant cyclin D1 (*Figure 2B, E, H, and K*). Then, we performed fluorescence-activated cell sorting (FACS) analysis and treated cells with nocodazole to synchronize the cell division cycle. We found that in HCT-116 cells stably transfected with the WT cyclin D1, forced expression of Keap1, DDB2, WSB2, or Rbx1 prevented the nocodazole-mediated G2/M block and subsequently resulted in accumulation of cells in G1 phase (*Figure 2—figure supplement 1*). In contrast, Keap1, DDB2, WSB2, or Rbx1 failed to induce efficient cell cycle arrest at the G1 phase in the mutant cyclin D1 (T286A) transfected cells (*Figure 2B, E, H, and K*). The results indicate that these cullin-associated ubiquitin ligases promoted cyclin D1 degradation and subsequently decreased cell cycle progression rate in a phosphorylation-dependent manner. Moreover, forced expression of Keap1, DDB2, WSB2, or Rbx1 also blocked DNA synthesis in WT cyclin D1 transfected cells but not mutant cyclin D1 (T286A) transfected cells (*Figure 2C, F, I, and L*). Consistent with these findings, we also observed that decreased phospho-Rb protein levels in the cells in which Keap1, DDB2, WSB2, or Rbx1 were co-transfected with WT cyclin D1 but not the mutant cyclin D1 (T286) (*Figure 2—figure supplement 2*). It was reported that Lysine 269 is essential for cyclin D1 ubiquitination (*Barbash et al., 2009*). To determine if mutation of Lysine 269 will affect cyclin D1 degradation induced by newly identified E3 ligases in the current study, WT or mutant cyclin D1 (K269R) expression plasmids were co-transfected with Keap1 or DDB2 expression plasmid into HEK293 cells. AMBRA1 expression plasmid, a

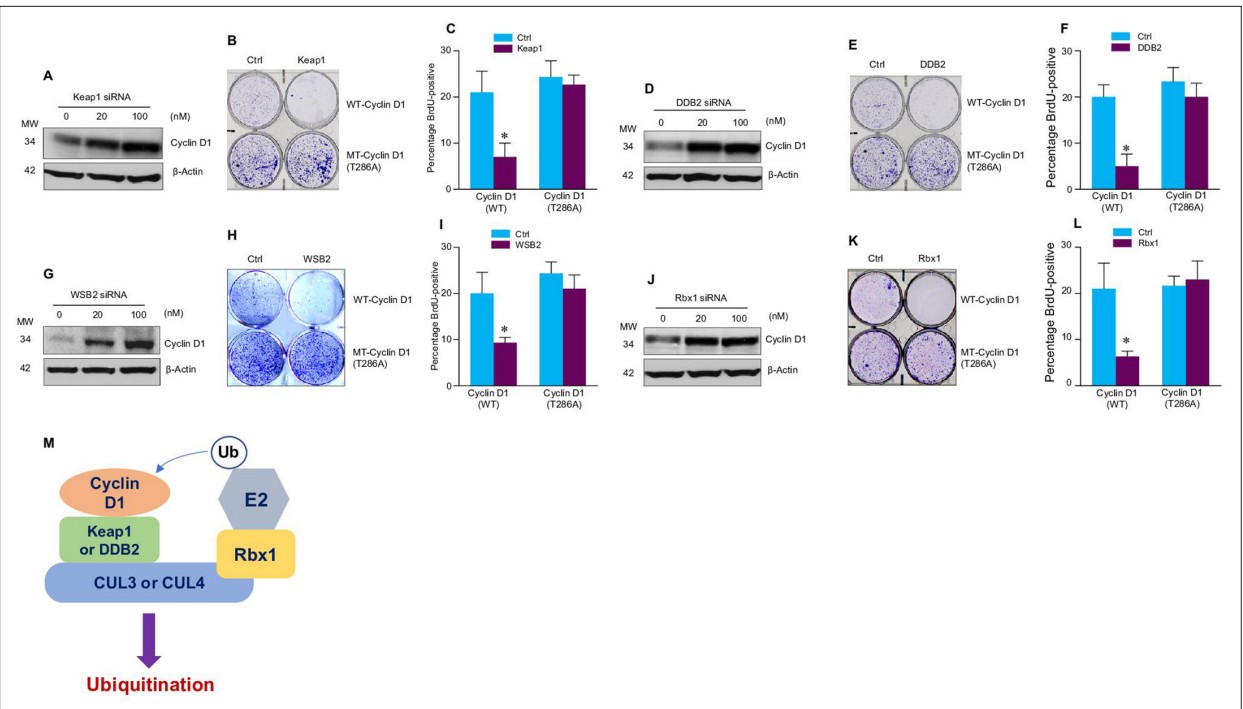

**Figure 2.** Cullin-associated E3 ligases affect cyclin D1 function. (**A, D, G, J**) Keap1, DDB2, WSB2, or Rbx1 siRNAs were transiently transfected into human colon cancer cell line HCT-116 cells. Endogenous cyclin D1 protein levels were detected by the western blot using the anti-cyclin D1 antibody. (**B, E, H, K**) Cell proliferation assay. Keap1, DDB2, WSB2, or Rbx1 expression plasmids were transfected into HCT-116 cells which were stably transfected with wild-type (WT) or mutant cyclin D1 (T286). The cells were stained with crystal violet 5 days after Keap1, DDB2, WSB2, or Rbx1 transfection. (**C, F, I, L**) Bromodeoxyuridine (BrdU) incorporation assay. Keap1, DDB2, WSB2, or Rbx1 expression plasmid was transiently transfected into HCT-116 cells which were stably transfected with WT or mutant cyclin D1 (T286). 4 hr before the harvest of cells, the cells were treated with BrdU (20 µM). 48 hr after the transfection, BrdU incorporation assays were performed (n=3). Data were presented as means ± SD of three independent experiments. Statistical analyses were performed using two-way ANOVA followed by the Tucky's post-hoc test, $*P<0.05$ (n=3). (**M**) A model showing that cullin-associated ubiquitin ligases are participated in the cyclin D1 proteolysis process.

The online version of this article includes the following source data and figure supplement(s) for figure 2:

**Source data 1.** Numerical data obtained during experiments represented in *Figure 2*.

**Source data 2.** Original western blot files for *Figure 2*.

**Figure supplement 1.** Fluorescence-activated cell sorting (FACS) analysis.

**Figure supplement 2.** Cullin-associated E3 ligases decreased phospho-Rb expression.

**Figure supplement 3.** Cullin-associated E3 ligases couldn't degrade K269R mutant cyclin D1.

**Figure supplement 4.** In vitro ubiquitination assay of cyclin D1 by Keap1, DDB2, WSB2, or AMBRA1.

well known E3 ligase targeting cyclin D1, was used as a control in this experiment. 48 hr after transfection, changes in cyclin D1 protein levels were detected by the western blot analysis. We found that expression of Keap1 or DDB2 in 293 cells reduced WT but not K269R mutant cyclin D1 protein levels (*Figure 2—figure supplement 3*). To determine if these newly identified E3 ligases directly interact cyclin D1, we performed in vitro ubiquitination assay and used AMBRA1 as a positive control. We found that cyclin D1 ubiquitination ladders were observed when Keap1 or DDB2 were added, suggesting that Keap1 and DDB2 could directly interact with cyclin D1. In contrast, WSB2 only has weak effect to directly interact with cyclin D1 (*Figure 2—figure supplement 4*).

Our findings indicate that cyclin D1 degradation is mediated by multiple E3 ligases, which are associated with different cullin proteins, including CUL2, -3, -4A, -4B, and -5. Several F-box proteins, Fbxw8, Fbxo4, Fbxo31, and AMBRA1, have previously been reported to mediate cyclin D1 degradation, and these three E3 ligases interact with cullin 1, 4, or 7 (*Chaikovsky et al., 2021*; *Kumar et al., 2005*; *Li et al., 2018*; *Maiani et al., 2021*; *Okabe et al., 2006*; *Santra et al., 2009*; *Simoneschi et al., 2021*). In our study, we found that Keap1 (works together with CUL3), DDB2 (works together with CUL4A and -4B), and WSB2 (works together with CUL2 and -5), which functions as the Ring

subunit binding protein interacting with the C-terminal domains of the cullins (*Petroski and Deshaies, 2005*), participated in the regulation of cyclin D1 proteolysis and affected the cell cycle progression (*Figure 2M*). The cullins interact with substrate receptor subunits to form ubiquitin ligase complex and mediate cyclin D1 degradation through a phosphorylation-dependent mechanism. This redundant cyclin D1 regulatory mechanism may function to ensure the normal G1-S phase transition and cell cycle progression in the cell.

Compared with transcriptional regulatory mechanism, post-translational regulation controls protein stability and plays a major role in the response to exogenous signals. It has been demonstrated that chemical reagents or γ-irradiation, which induce DNA damage, result in decreased cyclin D1 protein stability (*Choo et al., 2009*). DDB2 functions as a decisive factor for cell fate after DNA damage. DDB2-deficient cells are resistant to apoptosis in response to a variety of DNA damaging agents and also undergo cell cycle arrest (*Stoyanova et al., 2009*). Our results raise the possibility that DDB2 may regulate cell cycle pace after DNA damage through regulation of cyclin D1 protein stability.

In addition, cyclin D1 is abnormally up-regulated in many different types of human cancers as a proto-oncogene, including breast, lung, oesophagus, bladder, and lymphoid cancers (*Gautschi et al., 2007*). Genetic lesions, such as gene amplification or mutations, can only account for a portion of the cases for the abnormal up-regulation of cyclin D1 in these cancers. Growing evidences have shown that defective regulation at the post-translational level may be an important reason resulting in the increased cyclin D1 stability. The half-life of cyclin D1 in normal cells is relatively short (~20 min) due to ubiquitin-proteasome degradation regulated by a number of E3 ligases. In the current study we found that silencing of these ubiquitin ligases resulted in increased stability and accumulation of cyclin D1 in human cancer cells leading to the stimulation of cell cycle progression. In fact, a number of these cullins-associated E3 ligases, such as Fbxo31, Keap1, DDB2, and the cullins, have been demonstrated to function as tumor suppressors or components of tumor suppressor complexes (*Fay et al., 2003*). These tumor suppressors may exert their functions through controlling the steady-state protein levels of cyclin D1. Loss-of-function mutations of these E3 ligases lead to spontaneous tumor development (*Ohta et al., 2008*).

## Materials and methods

### Western blotting, immunoprecipitation, and ubiquitination assay

Western blotting and immunoprecipitation (IP) were performed as previously described (*Zhang et al., 2009*). The interaction between endogenous cyclin D1 and cullins-associated ligases subunits was determined in HEK293 cells. Proteasome inhibitor MG132 (10 μM) (Sigma, St. Louis, MO, USA) was added to the cell culture 4 hr before cells were harvested for IP or ubiquitination assay. For in vivo ubiquitination assay, HEK293 cells were co-transfected with plasmids expressing HA-cyclin D1 or HA-cyclin D1 (T286A) and cullins-associated ligases subunits constructs. Polyubiquitinated cyclin D1 was detected by co-IP using anti-ubiquitin antibody conjugated beads, followed by immunoblotting with an anti-HA antibody for cyclin D1. Blots were probed with the following antibodies:

| Antibodies | Source | Identifier |
|---|---|---|
| Cyclin D1 | Abcam | RRID: AB_2750906 |
| Cyclin B1 | Abcam | RRID: AB_731779 |
| Cyclin A | Santa Cruz Biotechnology | RRID: AB_627334 |
| Phospho-cyclin D1 (Thr286) | Cell Signaling Technology | RRID: AB_2070561 |
| Myc | Sigma-Aldrich | RRID: AB_309725 |
| HA | Santa Cruz Biotechnology | RRID: AB_2894930 |
| Flag | Sigma-Aldrich | RRID: AB_439687 |
| Ubiquitin | Abcam | RRID: AB_2801561 |
| CUL4A | Cell Signaling Technology | RRID: AB_2086563 |
| CUL4B | Proteintech | RRID: AB_2086699 |

*Continued on next page*

*Continued*

| Antibodies | Source | Identifier |
|---|---|---|
| WSB2 | Proteintech | RRID: AB_2216206 |
| Phospho-Rb (Ser780) | Cell Signaling Technology | RRID: AB_10950972 |

## Cell cycle analysis

FACS analysis and bromodeoxyuridine (BrdU) incorporation assay were performed as described before (*Santra et al., 2009*). For FACS analysis, HCT-116 cells were stably transfected with WT cyclin D1 or mutant (MT) cyclin D1 (T286A). The cells were then transfected with Keap1, WSB2, DDB2, or Rbx1. The cells were harvested 48 hr after the transfection. 16 hr before the harvest, the cells were treated 8 µg/ml of nocodazole (Sigma) for the final 16 hr. As for the non-nocodazole treatment experiments, HCT-116 cells were transiently transfected with siRNA for Keap1, DDB2, WSB2, or Rbx1. 48 hr after the transfection, the cells were synchronized at the G0 phase through serum starvation for over 16 hr. The cells were then stained with propidium iodide (50 µg/ml) at 37°C for 1 hr. FACS samples were analyzed with a FACSCanto Flow Cytometry System (BD Biosciences). And the data were analyzed using FlowJo 7.6 software according to the manufacturer's instruction. For labeling with BrdU, HCT-116 cells were stably transfected with WT cyclin D1 or MT cyclin D1 (T286A). The cells were then transiently transfected with Keap1, WSB2, DDB2, or Rbx1. The cells were harvested 48 hr after the transfection. Four hr before the harvest, the cells were treated with BrdU (20 µM).

## Cell culture and transfection

Human colon cancer HCT-116 cells (ATCC, RRID: CVCL_0291) and human embryonic kidney 293 (HEK293) cells (ATCC, RRID: CVCL_0045) were cultured in Dulbecco's modified Eagle's medium supplemented with 10% fetal calf serum at 37°C under 5% $CO_2$. The cell lines were tested for mycoplasma-free status before they were used. HCT-116 cells expressing WT HA-cyclin D1 or MT HA-cyclin D1 (T286A) were generated by transient transfection using Lipofectamine 2000 (Invitrogen, Carlsbad, CA, USA). Then transfected colonies were selected in the presence of G418 (1000 µg/ml for HCT-116 cells). DNA plasmids were transiently transfected into HEK293 cells in 6 cm culture dishes using Lipofectamine 2000. Empty vector was used to keep the total amount of transfected DNA plasmid constant in each group in all experiments. Flag-EGFP plasmid was co-transfected as an internal control to evaluate transfection efficiency. Western blotting and IP assays were performed 24 hr after transfection.

## Plasmids

Myc-CUL3 and FBXW8 were generously provided by Dr. Yue Xiong and Dr. Osamu Tetsu, respectively. Plasmids expressing WT HA-cyclin D1 and MT HA-cyclin D1 (T286A), CUL2, -4A, -4B, -5, and -7, Keap1 and Keap1 delta BTB, DDB2, Rbx1 were purchased from Addgene. The plasmid pCMV6-WSB2 (NM_018639.3) was purchased from OriGene. Loss-of-function MT DDB2 (WDΔ238–278), WSB2 (SOCSΔ364–400), and Rbx1 (C53A/C56A) were generated using site directed mutagenesis kit (Agilent, CA, USA). All constructs were confirmed by sequencing.

## In vivo protein decay assay

Cells were seeded in 15 cm culture dishes, cyclin D1 and equal amounts of siRNAs for Keap1, WSB2, DDB2, Rbx1, or control siRNA were transfected, respectively. 24 hr after transfection, cells were trypsinized and split into five 10 cm dishes. 12 hr after recovery, cells were cultured in regular medium with 80 µg/ml cycloheximide (Calbiochem, La Jolla, CA, USA), for 0, 30, 60, 120, and 300 min before harvesting. Western blotting was performed to detect the decay of cyclin D1 proteins.

## Cullin-associated E3 ligases subunits siRNA library screening

462 unique siRNAs targeting each of 156 genes were coated in 96-well plates (silencer Custom siRNA library, Ambion, Austin, TX, USA). 6xE2F luciferase reporter, cyclin D1 plasmids were co-transfected

with the siRNA library into NIH3T3 cells. 24 hr after the transfection, the cell lysates were collected, and luciferase activity was measured using a Promega Dual Luciferase reporter assay kit (Promega, Madison, WI, USA).

### Real-time PCR

Cell samples were immediately posited in 1 ml TRIZOL (Invitrogen) Reagent after taking out from –80°C refrigerator, and further processed with TissueLyser for RNA extraction. Total cellular RNA was extracted by the TRIZOL Reagent according to the supplier's instructions. cDNAs of the samples were synthesized with RevertAid First Strand cDNA synthesis Kit (Thermo Fisher). And real-time PCR amplification of the cDNAs were performed with SYBR Premix Ex Taq (TAKARA) kit in ABI 7500 real-time PCR system, with following specific primers: cyclin D1 5'-CCGTCCATGCGGAAGATC-3' (upper primer) and 5'-GAAGACCTCCTCCTCGCACT-3' (lower primer); GAPDH, 5'-GAAGGTGAAGGT CGGAGT-3' (upper primer) and 5'-GAAGATGGTGATGGGATTTC-3 (lower primer). As for the real-time PCR in the gene knockdown experiments, the primers used in this study were as follows: *cyclin D1*, 5'-CGTGGCCTCTAAGATGAAGG (upper primer) and 3'-CTGGCATTTTGGAGAGGAAG (lower primer); *Cul1*, 5'-AATGCCCTGGTAATGTCTGC (upper primer) and 3'-GTCACAGTATCGAGCCAGCA (lower primer); *Cul2*, 5'-CTTACTCCGTGCTGTGTCCA (upper primer) and 3'-GCCTTATCCAACGCAC TCAT (lower primer); *Cul3*, 5'-TCCAGGGCTTATTGGATCTG (upper primer) and 3'-GCCCTTTGACTC CCTTTTTC (lower primer); *Cul4A*, 5'-AAAGAAGCCACAGACGAGGA (upper primer) and 3'-ATGT CCCTGAACATGCCTTC (lower primer); *Cul4B*, 5'-CGCCTGTTAGTCGGAAAGAG (upper primer) and 3'-TTCCCGGAACATTCTGATTC (lower primer); *Cul5*, 5'-TGCAGTCTGTCTTTGGGATG (upper primer) and 3'-TATTGCTGCCCTGTTTACCC (lower primer); *Cul7*, 5'-TAGAATTGGCCCAGGACTTG (upper primer) and 3'-GCGTCTAGCAGGAGGACATC (lower primer); *β-actin*, 5'-GGACTTCGAGCA AGAGATGG (upper primer) and 3'-AGCACTGTGTTGGCGTACAG (lower primer). The PCR conditions included a denaturation step at 95°C for 5 min, followed by 35 cycles of denaturation at 95°C for 10 s, annealing at 58°C for 15 s, and extension at 72°C for 10 s. Detection of the fluorescent product was carried out at the end of the 72°C extension period. The PCR products were subjected to a melting curve analysis, and the data were analyzed and quantified with the Rotor-Gene analysis software. Dynamic tube normalization and noise slope correction were used to remove background fluorescence. Each sample was tested at least in triplicate and repeated using three independent cell preparations.

### Luciferase and real-time PCR assays

The plasmids of reporter constructs were co-transfected with cullins and cyclin D1 expression plasmid into HEK293 cells. 24 hr after transfection, the cell lysates were then collected, and luciferase activity was measured using a Promega Dual Luciferase reporter assay kit.

### In vitro ubiquitination assay

The in vitro ubiquitination assay was performed according to the manufacturer's recommendations (ab139467, Abcam, USA). The cyclin D1 fusion protein (P05317, Solarbio, China) expressed in *Escherichia coli* was incubated at 30°C for the 30 min in the presence of E1, E2, ATP, Ub, and E3 ligase recombinant proteins including Keap1, DDB2, WSB2, RBX1, and AMBRA1 (Abnova, China). Samples were resolved by 8% SDS-PAGE and subjected to immunoblot analysis with the anti-cyclin D1 antibody.

### Statistics

Statistical comparison between two groups was performed using unpaired Student's *t*-test and the two-way ANOVA followed by the Tucky's post-hoc test (n=3). $p < 0.05$ was considered significant and is denoted in the figures.

## Acknowledgements

This work was supported by the National Key Research and Development Program of China (2021YFB3800800) to LT and DC. This project was supported by the National Natural Science Foundation of China (NSFC) grants 82030067, 82161160342, 82250710174, and 82172397 to DC and LT.

# Additional information

### Competing interests

Di Chen: Reviewing editor, *eLife*. The other authors declare that no competing interests exist.

### Funding

| Funder | Grant reference number | Author |
|---|---|---|
| National Key Research and Development Program of China | 2021YFB3800800 | Liping Tong<br>Di Chen |
| National Natural Science Foundation of China | 82030067 | Di Chen |
| National Natural Science Foundation of China | 82161160342 | Di Chen |
| National Natural Science Foundation of China | 82250710174 | Di Chen |
| National Natural Science Foundation of China | 82172397 | Liping Tong |

The funders had no role in study design, data collection and interpretation, or the decision to submit the work for publication.

### Author contributions

Ke Lu, Ming Zhang, Data curation, Formal analysis, Methodology; Guizheng Wei, Investigation; Guozhi Xiao, Formal analysis, Writing - review and editing; Liping Tong, Formal analysis, Project administration, Writing - review and editing; Di Chen, Conceptualization, Funding acquisition, Writing - original draft, Project administration, Writing - review and editing

### Author ORCIDs

Ke Lu https://orcid.org/0000-0002-1641-4748
Guizheng Wei http://orcid.org/0009-0006-0479-5076
Guozhi Xiao http://orcid.org/0000-0002-4269-2450
Di Chen http://orcid.org/0000-0002-4258-3457

### Decision letter and Author response

Decision letter https://doi.org/10.7554/eLife.80327.sa1
Author response https://doi.org/10.7554/eLife.80327.sa2

# Additional files

### Supplementary files
- Transparent reporting form

### Data availability

All data generated or analysed during this study are included in the manuscript and supporting files. Source data files have been provided for Figures 1, Figure 2 and the figure supplements.

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
