## [Editor Report]

This fundamental study advances our understanding of the role of cullin-associated E3 ligases in regulating cyclin D1 protein stability. The evidence supporting the conclusion is compelling, from siRNA screens and ectopic expression approaches. This paper is of potential interest for cell biologists studying the mechanisms of protein posttranslational modification.

---

## [Decision Letter]

**Decision letter after peer review:**

Thank you for submitting your article "Multiple Cullin-Associated E3 Ligases Regulate Cyclin D1 Protein Stability" for consideration by *eLife*. Your article has been reviewed by 3 peer reviewers, and the evaluation has been overseen by a Reviewing Editor and Mone Zaidi as the Senior Editor. The following individual involved in the review of your submission has agreed to reveal their identity: Chao Xie (Reviewer #1).

Essential revisions:

This interesting set of data provides new understanding of the role of cullin-associated E3 ligases in regulating cyclin D1 protein stability. The data within the manuscript largely support most conclusions. However, there are major concerns that need to be addressed.

1) The exact mechanism of how cyclin D1 is ubiquitinated and degraded is incomplete. It is important for the authors to show direct interactions and in vitro ubiquitylation assays. It is also important to put the findings in a coherent model that discusses the existing literature showing that CRL4-AMBRA1 is the major E3 ligase under most tested conditions.

2) It is necessary for the authors to include appropriate controls for the co-IP and CHX chase experiments.

*Reviewer #2 (Recommendations for the authors):*

The lack of identified lysine residue mutant within cyclin D1 that is deficient for ubiquitination is considered a significant weakness for the manuscript.

The in vitro ubiquitination assays of cyclin D1 by different combinations of cullin-E3 ligases will further strengthen the major claim of the manuscript.

*Reviewer #3 (Recommendations for the authors):*

1) One major oversight of the study has been not to include the studies describing CRL4-AMBRA1 as a major E3 ligase degrading cyclin D (1-3) (Simoneschi et al., Nature 2021; Maiani et al., Nature 2021; Chaikovsky et al., Nature 2021). These studies provide convincing evidence that CUL4 and AMBRA1, but not other cullins or substrate adaptors regulate endogenous cyclin D levels in several cell lines including RPE-1, U2OS, and HCT-116 cells, and show functional relevance during human development and cancer. Some of the results presented In these studies contradict the results presented in this short report (e.g. Simoneschi et al. show in Figure 1 that knockdown of only CUL4A/B and AMBRA1, but no other cullin or substrate adaptor, including DDB2, increase cyclin D1 levels in HCT-116 cells). Thus, the authors need to mention these previous findings in their introduction. This would obviously change the scope of their study and require them to address the discrepancies between these previous and their findings and also provide experimental evidence for when and how CRLs other than CRL4-AMBRA1 is important. This would ultimately result in a completely new study.

2) Overstatements:

– The authors claim that no other CRLs than CRL1/7 with substrate adaptors FBXW8, FBXO4, and FBXO31 has been implicated in Cyclin D degradation, which is not the case (see above).

– In the abstract they also claim that KEAP1, DDB2, and WSB2 directly interact with cyclin D1, something they did not test. They only provide evidence that these proteins, when ectopically expressed, can be in a complex with cyclin D1 in cells. These co-IP experiments also lack important controls (see below point 4). Direct interaction can only be claimed by in vitro reconstitution with recombinantly purified proteins.

– Throughout the manuscript effects with RBX1 are presented as novel findings; however, RBX1 is an integral part of all CRLs and thus should be presented as a positive control

3) Another important issue that arises from this study is how different CRLs would recognize cyclin D structurally. The authors draw a model (Figure 2M), in which all four different types of CRLs recognize cyclin D by the same mechanism using different substrate adaptors. This would be rather surprising, given that different substrate adaptors are generally thought to use unique manners to bind specific sets of substrates. Given that the authors do no provide evidence for direct interaction and ubiquitylation, It is equally conceivable that the effects they are observing are occurring through indirect mechanisms and unknown substrates (e.g. AMBRA1).

4) In addition to these major conceptual weaknesses, there are also missing controls for several experiments. To name a few: the anti-substrate adaptor co-IP experiments (Figure 1A-C) are lacking loading controls that show that the same amount of substrate adaptor was enriched in the IP. The ubiquitylation assays (Figure 1E-H) are missing controls that show that the same amount of ubiquitin was immunoprecipitated in each condition. The CHX assays (Figure 1M) need to be quantified and half-lives should be determined.

---

## [Author Response]

Essential revisions:This interesting set of data provides new understanding of the role of cullin-associated E3 ligases in regulating cyclin D1 protein stability. The data within the manuscript largely support most conclusions. However, there are major concerns that need to be addressed.1) The exact mechanism of how cyclin D1 is ubiquitinated and degraded is incomplete. It is important for the authors to show direct interactions and in vitro ubiquitylation assays. It is also important to put the findings in a coherent model that discusses the existing literature showing that CRL4-AMBRA1 is the major E3 ligase under most tested conditions.

In the revised manuscript we have performed in vitro ubiquitination assays and used AMBRA1 as a positive control in this assay. We found that Keap1, DDB2, and WSB2 can induce cyclin D1 ubiquitination. Especially, Keap1-induced cyclin D1 ubiquitination ladder is similar to AMBRA1-induced cyclin D1 ubiquitination ladder. In contrast, no clear ubiquitination ladder was observed in Rbx1 group (Figure 2—figure supplement 4). In addition, we have also added those key publications about the role of CRL4-AMBRA1 in cyclin D1 degradation in the revised manuscript as the reviewer suggested.

2) It is necessary for the authors to include appropriate controls for the co-IP and CHX chase experiments.

We have added ubiquitin input controls for co-IP (Figure 1E-H) and quantified the half-lives for CHX assays (Figure 1N).

Reviewer #2 (Recommendations for the authors):The lack of identified lysine residue mutant within cyclin D1 that is deficient for ubiquitination is considered a significant weakness for the manuscript.

It was reported that Lysine 269 is essential for cyclin D1 ubiquitination (Barbash et al., 2009). WT or mutant cyclin D1 (K269R) expression plasmids were co-transfected with Keap1, DDB2, or AMBRA1 expression plasmid into HEK293 cells. 48 hours after transfection, changes in cyclin D1 protein levels were detected by the western blot analysis. We found that expression of WT cyclin D1, but not K269R mutant cyclin D1, was decreased in HEK293 cells transfected with Keap1, DDB2, or AMBRA1 plasmid, suggesting that Lysine 269 is essential for cyclin D1 ubiquitination.

The in vitro ubiquitination assays of cyclin D1 by different combinations of cullin-E3 ligases will further strengthen the major claim of the manuscript.

We have performed in vitro ubiquitination assay as the reviewer suggested. The results demonstrated that Keap1, DDB2, or WSB2 can induce cyclin D1 ubiquitination. Especially, Keap1 induced cyclin D1 ubiquitination and formed ubiquitination ladder similar to AMBRA1-induced cyclin D1 ubiquitination ladder. In contrast, no clear ubiquitination ladder was observed in Rbx1 group (Figure S16).

Reviewer #3 (Recommendations for the authors):1) One major oversight of the study has been not to include the studies describing CRL4-AMBRA1 as a major E3 ligase degrading cyclin D (1-3) (Simoneschi et al., Nature 2021; Maiani et al., Nature 2021; Chaikovsky et al., Nature 2021). These studies provide convincing evidence that CUL4 and AMBRA1, but not other cullins or substrate adaptors regulate endogenous cyclin D levels in several cell lines including RPE-1, U2OS, and HCT-116 cells, and show functional relevance during human development and cancer. Some of the results presented In these studies contradict the results presented in this short report (e.g. Simoneschi et al. show in Figure 1 that knockdown of only CUL4A/B and AMBRA1, but no other cullin or substrate adaptor, including DDB2, increase cyclin D1 levels in HCT-116 cells). Thus, the authors need to mention these previous findings in their introduction. This would obviously change the scope of their study and require them to address the discrepancies between these previous and their findings and also provide experimental evidence for when and how CRLs other than CRL4-AMBRA1 is important. This would ultimately result in a completely new study.

We agree that those findings about CRL4-AMBRA1 in cyclin D1 degradation provide most critical information in the regulation of steady-state protein levels of cyclin D1 in the cell. In addition, we have used CRL4-AMBRA1 as a positive control in our studies during revision of our manuscript.

2) Overstatements:– The authors claim that no other CRLs than CRL1/7 with substrate adaptors FBXW8, FBXO4, and FBXO31 has been implicated in Cyclin D degradation, which is not the case (see above).

We have added the information about critical role of CRL4-AMBRA1 in regulation of cyclin D1 protein stability in the revised manuscript.

– In the abstract they also claim that KEAP1, DDB2, and WSB2 directly interact with cyclin D1, something they did not test. They only provide evidence that these proteins, when ectopically expressed, can be in a complex with cyclin D1 in cells. These co-IP experiments also lack important controls (see below point 4). Direct interaction can only be claimed by in vitro reconstitution with recombinantly purified proteins.

We have performed experiments to determine if Keap1, DDB2, or WSB2 could directly interact with cyclin D1 and induce cyclin D1 ubiquitination and we found that Keap1, DDB2, and WSB2 can induce cyclin D1 ubiquitination. Especially, Keap1 induced cyclin D1 ubiquitination and formed ubiquitination ladder similar to AMBRA1-induced cyclin D1 ubiquitination ladder. In contrast, no clear ubiquitination ladder was observed in Rbx1 group (Figure 2—figure supplement 4). We also added input control in the co-IP assays as the reviewer suggested.

– Throughout the manuscript effects with RBX1 are presented as novel findings; however, RBX1 is an integral part of all CRLs and thus should be presented as a positive control

We have presented Rbx1 as a control in the revised manuscript as the reviewer suggested.

3) Another important issue that arises from this study is how different CRLs would recognize cyclin D structurally. The authors draw a model (Figure 2M), in which all four different types of CRLs recognize cyclin D by the same mechanism using different substrate adaptors. This would be rather surprising, given that different substrate adaptors are generally thought to use unique manners to bind specific sets of substrates. Given that the authors do no provide evidence for direct interaction and ubiquitylation, It is equally conceivable that the effects they are observing are occurring through indirect mechanisms and unknown substrates (e.g. AMBRA1).

In the revised manuscript, we have performed in vitro ubiquitination assay as the reviewer suggested. Together, with co-IP assay and in vitro ubiquitination assay, our findings suggest that some E3 ligase that we have identified could directly interact with cyclin D1 and some others probably affect cyclin D1 protein stability by indirect mechanisms.

4) In addition to these major conceptual weaknesses, there are also missing controls for several experiments. To name a few: the anti-substrate adaptor co-IP experiments (Figure 1A-C) are lacking loading controls that show that the same amount of substrate adaptor was enriched in the IP. The ubiquitylation assays (Figure 1E-H) are missing controls that show that the same amount of ubiquitin was immunoprecipitated in each condition. The CHX assays (Figure 1M) need to be quantified and half-lives should be determined.

We have added proper controls for co-IP assays and ubiquitination assays and quantified the half-lives for CHX assays in the revised manuscript as the reviewer suggested.

Reference

Barbash, O., Egan, E., Pontano, L.L., Kosak, J., and Diehl, J.A. (2009). Lysine 269 is essential for cyclin D1 ubiquitylation by the SCF(Fbx4/alphaB-crystallin) ligase and subsequent proteasome-dependent degradation. Oncogene *28*, 4317-4325.